Systematic review and meta-analysis on the effect of continuous subjective tinnitus on attention and habituation

Vasudevan Harini 1
Ganapathy Kanaka 1
Palaniswamy Hari Prakash hari.prakash@manipal.edu 1
Searchfield Grant 2
Rajashekhar Bellur 1
1 Department of Speech and Hearing, Manipal College of Health Professions (MCHP), Manipal Academy of Higher Education (MAHE) , Manipal , karnataka , India
2 Faculty of Medical and Health Sciences, University of Auckland , Auckland , New Zealand
Andersson Gerhard
Electronic publication date: 2021 Nov 26
Publication date: 2021
Volume: 9
Electronic Location ID: e12340
Received 2021 Apr 5; Accepted 2021 Sep 28
Copyright: ©2021 Vasudevan et al.
Copyright year: 2021
Copyright holder: Vasudevan et al.
License: This is an open access article distributed under the terms of the Creative Commons Attribution License, which permits unrestricted use, distribution, reproduction and adaptation in any medium and for any purpose provided that it is properly attributed. For attribution, the original author(s), title, publication source (PeerJ) and either DOI or URL of the article must be cited.
License URL: https://creativecommons.org/licenses/by/4.0/

Keywords: Tinnitus, Attention, Habituation, Systematic review

Funding: The authors received no funding for this work.

==============================
Background

Attention and habituation are two domains known to play key roles in the perception and maintenance of tinnitus. The heterogeneous nature of tinnitus and the methodologies adopted by various studies make it difficult to generalize findings. The current review aims at assessing and synthesizing evidence on the possible roles of attention and habituation in continuous subjective tinnitus.

Methodology

The literature search included five databases (PubMed, Scopus, Web of Sciences, CINAHL and ProQuest) that resulted in 1,293 articles, published by July 2019. Studies on attention and/or habituation in individuals with tinnitus using either behavioural or electrophysiological tests were included in the review after a quality assessment.

Results

Seventeen studies on attention in tinnitus were included in the narrative synthesis. Two meta-analyses were performed to assess the role of attention in tinnitus using a behavioural methodology (z = 4.06; p < 0.0001) and P300 amplitude (z = 2.70; p = 0.007) with 531 participants. With respect to habituation, the review indicates the lack of quality articles for habituation inclusion in the narrative synthesis.

Conclusions

The review highlights that selective domains of attention were consistently impaired in individuals with tinnitus. Habituation, on the other hand, needs further exploration.

Introduction

Tinnitus is the conscious awareness of a tonal or composite noise for which there is no identifiable corresponding external acoustic source (De Ridder et al., 2021). The pathophysiology of tinnitus is complex involving various cortical and subcortical systems with primary damage to the auditory periphery (Galazyuk, Wenstrup & Hamid, 2012). It manifests either continuously, or in an intermittent form, and is experienced by about 10–15% of the world’s population based on various epidemiological studies (Baigi et al., 2011; Gopinath et al., 2010; Hasson et al., 2011; Hasson et al., 2010; Michikawa et al., 2010; Park et al., 2014; Shargorodsky, Curhan & Farwell, 2010). However, only a portion of individuals having tinnitus find it disturbing with a recent suggestion that this more disabling tinnitus be defined as Tinnitus Disorder (De Ridder et al., 2021). A contributing factor to Tinnitus Disorder may be the attention focused on tinnitus and an individual’s ability to become habituated to the tinnitus sound.

Attention, a major domain under cognition, is the process of allocating cognitive resources to focus on information processing. Attention includes sub-domains like alerting, orienting, sustained attention, selective attention, divided attention and executive attention. Active or passive attention towards the tinnitus could drive the cognitive resources away from the primary task that is being performed. In addition, it also makes habituation to tinnitus difficult resulting in decompensating or chronic tinnitus.

Over the years, different types of attention have been studied using behavioural tests like the Stroop task, vigilance task, divided and sustained attention tasks, flanker’s paradigm or using electrophysiological measures like P300 and Mismatch Negativity (MMN) for studying active and passive attention, respectively. MMN reflects the pre-attentive process to discriminate the stimulus based on their perceptual characteristics (Näätänen, 2001), whereas P300 reflects a higher-level attentional resource and working memory update during the process of perceptual discrimination (Polich, 2012). Although a majority of literature supports that attention is affected in individuals with tinnitus (Andersson et al., 2000; Asadpour et al., 2018; Cuny et al., 2004; Dos Santos Filha & Matas, 2010; Gabr, Abd El-Hay & Badawy, 2011; Heeren et al., 2014; Hong et al., 2016; Jackson, Coyne & Clough, 2014; Li et al., 2016; Lima et al., 2020; Mahmoudian et al., 2013; Mannarelli et al., 2017; Mohebbi et al., 2019; Stevens et al., 2007; Wang et al., 2018), some studies do not (Davies, McKenna & Hallam, 1995; Elmorsy & Abdeltawwab, 2013; Hallam, McKenna & Shurlock, 2004; Houdayer et al., 2015; Najafi & Rouzbahani, 2020; Shiraishi et al., 1991; Waechter & Brännström, 2015). The inconsistency in the literature results in ambiguity as to the true role of attention in tinnitus.

Habituation is a form of learning wherein the response to a stimulus that has been repeated or presented for a long time decreases or ceases (Bouton, 2007). Habituation or passive extinction is essential for the brain to perform multiple tasks simultaneously. The brain constantly updates its schema based on the incoming sensory input. Repeated presentation of stimulus is considered as predictable by the brain and as a result, the perceptual salience allocated to it is less (Durai, O’Keeffe & Searchfield, 2018). Habituation is a core premise of several important models of tinnitus including the neurophysiological model, (Jastreboff, 1990), habituation model (Hallam, Rachman & Hinchcliffe, 1984) and therapies like tinnitus retraining therapy, (Jastreboff & Jastreboff, 2000); habituation therapy (Andersson & McKenna, 1998; Coles & Hallam, 1987) and guided therapy (Slater, Terry & Davis, 1987).

Jastreboff’s tinnitus model suggests that persons with tinnitus, but without associating any negative emotions, can become habituated to the tinnitus (Jastreboff, Gray & Gold, 1996). The perception of tinnitus gets enhanced only when a person is consciously paying attention to it. Until a negative emotion gets tagged to this sound, the limbic and autonomic nervous system(ANS) co-activation with the tinnitus sound is limited. However, when paired the ANS gets conditioned to the tinnitus signal and negative reactions like fear and annoyance accompany tinnitus, creating a “vicious cycle”. The presence of this negative reinforcement from the associated systems makes it difficult for habituation to occur.

Habituation to an external sound may be different from habituation to the internal sounds like tinnitus. Since there are no standardized test to study habituation to tinnitus, measures like P50 can be used to evaluate the sensory gating, thereby indirectly assessing habituation. “Sensory gating” is a phenomenon where the brain automatically analyses the incoming stream of information based on its salience to determine the weight that must be given to the stimulus. P50 is an electrophysiological measure that is used widely to evaluate the sensory gating mechanism at the thalamo-cortical level using a paired click paradigm. The redundant or the second click in the paradigm is given less importance, which is observed as reduced P50 amplitude for the redundant stimuli. Individuals having schizophrenia are reported have reduced sensory gating abilities (Shen et al., 2020). With respect to sensory gating in tinnitus population, there are only a handful of studies that have assessed sensory gating in tinnitus experimentally suggesting affected sensory gating in individuals with tinnitus (Campbell, Bean & LaBrec, 2018; Campbell et al., 2019) while others suggested it to be normal (Dornhoffer et al., 2006).

Attention and habituation appear to be two important domains in the perception and maintenance of tinnitus. The current review differs from the existing reviews (Cardon et al., 2020; Clarke et al., 2020) in such a way that, we explore the effect of continuous tinnitus on attention and habituation solely instead of cognition as a whole. Assessing these two specifically in individuals with tinnitus is crucial to understand the roles these domains play, the selection and development of appropriate therapies. In addition, the existing reviews have not addressed the behavioral and electrophysiological indices of attention together nor have, they assessed attention in tinnitus by controlling confounders like hearing loss. Furthermore, existing reviews have included studies with pulsatile tinnitus making the group heterogenous. The current review aims to overcome the above by exploring the sustained effect of continuous and subjective tinnitus on attention and habituation using both behavioral and electrophysiological measures in adults.

Survey methodology

The current review protocol was registered and approved by PROSPERO (CRD42019127207).

Keyword Build

Using the Cochrane library, Medical term [MeSH] search engine, all necessary terms for keyword “Tinnitus” along with the “Attention” or “Habituation were identified, and the search string was built using appropriate Boolean operators. (Key words: Tinnitus [MeSH], AND “P300”, OR ”auditory P3”, OR P3, OR “cognitive potential”, OR “stroop task”, OR “Attentional network task”, OR “ANT”, OR “Attentional network test”, OR “flankers”, OR “flankers paradigm ”, OR “flankers test”, OR “event related potentials”, OR “event related potential”, OR ERP, OR “ERPs”, OR “cortical auditory evoked potentials” OR “cortical auditory evoked potential” OR “CAEPs” OR “CAEP” OR ”stroop test”, OR attention, OR “selective attention”, OR “auditory selective attention” OR “sustained attention”, OR “executive attention”, OR “alerting attention” OR “focussed attention”, OR habituation, OR “thalamo cortical habituation”, OR “cortical habituation”, OR “sensory gating”, OR “auditory gating”, OR “mismatch negativity”, OR “MMN”, OR “P50”).

Search strategy

Thirty-six keywords were used to search five major databases including PubMed, Scopus, Web of Sciences, CINAHL and ProQuest. There were no restrictions pertaining to language. The search was predominantly run through the title and/or abstract of all articles published till the 25th of July 2019. 1293 articles were retained, after removing duplicates (n = 978) in Covidence software.

Title and abstract screening/selection process

Two independent reviewers (Reviewer 1, HV and Reviewer 2, KG) screened the articles through title and abstract. Conflicts that arose were resolved by reviewer 3 (HP). On this initial screening, 102 articles that assessed attention and/or habituation in individuals with tinnitus qualified for full-text screening.

Full-text screening

A similar screening was carried out by two reviewers (HV and KG) independently with conflict resolution by reviewer 3, HP.

Inclusion-exclusion criteria

• Studies that assessed continuous subjective tinnitus on the adult population (18 years and above) that addressed attention and/or habituation using either behavioural or electrophysiological measures were included.

• Study types including observational, cross-sectional studies, case-control or cohort studies were included for full-text screening.

• Articles that addressed only simulated tinnitus, pulsatile tinnitus, qualitative study on an individual’s experience with tinnitus, treatment (controlled trials and RCTs), and systematic reviews were eliminated

• Articles in languages other than English were eliminated.

Based on full-text screening, 33 articles were found suitable for the narrative review.

Risk of bias (ROB) analysis

Quality assessment of 33 articles was carried out by two independent reviewers (Reviewer 4, GS and Reviewer 5, BR) and conflicts resolved by reviewer 3 (HP). To screen the risk of bias, appropriate questions from CASP (Critical Appraisal Skilled Programme for case-control studies) were considered. The studies were appraised based on whether they utilized a thorough and appropriate methodology, the meaning and credibility of study findings, and their relevance. Based on the above, the reviewers were asked to rate the risk of bias of the articles on a 5-point scale from very high risk to very low risk. Based on collective inputs from the reviewers, 16 articles were rejected. The ROB analysis and the reasons for rejection are shown in the supplementary file. Finally, 17 studies with low to moderate risk were included in the narrative synthesis. The complete process from searching for articles to those included in the review is represented in the PRISMA chart (Fig. 1).

Figure 1 PRISMA flowchart.

Data extraction

Data extraction was carried out in an excel spreadsheet by two independent reviewers (HV and KG). The data extracted included the following: age range of participants, gender, number of participants in each group, place of study, matching of controls, tinnitus pitch and loudness, tinnitus laterality, duration, severity, history of previous treatment, residual inhibition information, scales used to assess tinnitus, participants hearing level, degree of hearing loss, screening for psychological characteristics, the behavioral or electrophysiological test performed with an elaborate method, outcome, and justification, stimulus modality, stimulus information like frequency, duration, intensity, inter-stimulus interval, the instrument used, channel information, pre-processing of data, statistical analysis, the main findings of the study with justification, possible treatment options for tinnitus and future directions. The extracted data were placed into different categories namely, general information, tinnitus characteristics, hearing acuity, psychological and psychiatric screening, the test used and outcomes, stimulus, and instrumentation information and the main results, discussion, and future direction.

Data synthesis

The extracted data was synthesized into a narrative form under various categories including age and gender, place of study, the tinnitus characteristics, hearing acuity, psychological and psychiatric factors, the overall test done with their outcomes, and the instrumentation used. For those articles where quantitative data were obtained a meta- analysis was performed.

Meta-analysis

Those articles with necessary quantitative data were synthesized into a meta-analysis using Review Manager (version 5.2). Two meta-analyses were performed to find the effect of tinnitus on attention. Firstly, using the reaction time in milliseconds provided by the behavioral studies and secondly with the P300 amplitude in microvolt from the electrophysiological studies. A random-effects meta-analysis was done using the standardized mean difference (SMD) for the behavioral studies and Mean Difference (MD) for the P300 studies between the tinnitus and control group with a 95% confidence interval. Further a subgroup analysis on the basis of hearing was conducted. Two random effects meta-analyses were conducted with those who have matched for hearing and those who have not matched. The results of the meta-analysis were evaluated based on the pooled evidence to calculate the overall effect (p-value).

Results

Out of the seventeen studies included, nine used behavioural tests to assess one or more types of attention; the other eight employed an electrophysiological paradigm to assess the same. With respect to habituation, no studies passed the risk of bias assessment to be included in the narrative synthesis.

Narrative synthesis

Age and gender

Matching age and gender in hearing research is one of the essential steps in case-control design to create a homogenous group. In the current review, it was found that, out of the seventeen studies, sixteen had controlled either age and/or gender. Most of the studies had matched for age except for Cuny et al. (2004) and Houdayer et al. (2015). Seven studies had included the geriatric population (60 years and older) (Andersson et al., 2000; Araneda, Deggouj & Renier, 2015; Heeren et al., 2014; Rossiter, Stevens & Walker, 2006; Shiraishi et al., 1991; Stevens et al., 2007; Trevis, McLachlan & Wilson, 2016). All the studies matched for gender except for five (Houdayer et al., 2015; Jackson, Coyne & Clough, 2014; Rossiter, Stevens & Walker, 2006; Shiraishi et al., 1991; Stevens et al., 2007).

Place of study

Eight of the studies were carried out in European countries (United Kingdom, Sweden, Belgium, Rome, Italy, France, and Spain), three in Australia, two each in Israel and Iran, one in Japan and one in Korea.

Tinnitus characteristics

Six of the seventeen studies provided tinnitus pitch match results while five matched the loudness of tinnitus. Most of the studies (n = 12) had included participants with both unilateral and bilateral tinnitus. All studies, except Shiraishi et al. (1991) and Cuny et al. (2004), had included tinnitus duration information. The duration of tinnitus ranged from 3 months to 7 years. The commonly used questionnaires were the Tinnitus Handicap Inventory, THI (Newman, Jacobson & Spitzer, 1996) (n = 7), Tinnitus Questionnaire, TQ (Hallam, Jakes & Hinchcliffe, 1988) (n = 6), Tinnitus Sample Case History Questionnaire, TCSHQ (Langguth et al., 2007), Subjective Tinnitus Severity Scale, STSS (Halford & Anderson, 1991), Tinnitus Reaction Questionnaire, TRQ (Wilson et al., 1991), Tinnitus Psychological Impact Questionnaire, QIPA (Philippot et al., 2012), and Tinnitus Severity and Symptom profile questionnaire (Barnea et al., 1990). However, Attias et al. (1993) and Shiraishi et al. (1991) did not report the use of any questionnaire. The tinnitus characteristics of the participants included in the review studies are depicted in Table 1.

Table 1 Tinnitus characteristics reported in the review.

Study
(no. of tinnitus participants)	Pitch	Loudness	Laterality	Previous treatment	Duration of tinnitus
(in months)	Severity	Scale used to measure	
Andersson et al. (2000)
( n= 23)	Mean
5.59 kHz	19dBSL (18)	B/L	Yes, 8/23 underwent	6.3 (4.1)	Severe	S-TQ	
Araneda, Deggouj & Renier (2015)
( n= 17)	0.25–8 kHz	NR	B/L	No	6 and above	Mild to Severe	TSCHQ, THI	
Attias et al. (1993)
( n= 12)	5–8 kHz	10–20 dBSL	B/L	NR	60 and above	NR	Not used	
Attias et al. (1996)
( n= 21)	NR	NR	B/L	NR	84 and above	Chronic	Tinnitus severity and symptom profile questionnaire	
Cuny et al. (2004)
( n= 20)	NR	NR	B/L	NR	NR	NR	STSS	
Heeren et al. (2014)
( n= 20)	NR	NR	B/L	No masker related treatment	6	NR	QIPA	
Hong et al. (2016)
( n= 14)	8 kHz	NR	B/L	No	3 and above	Range varied	TQ and THI	
Houdayer et al. (2015)
( n= 17)	4, 6, & 8 kHz	6.41 (2.96)
dBSL	U/L	NR	22	Chronic	THI	
Jackson, Coyne & Clough (2014)
( n= 33)	NR	NR	NR	No	Not above12	Low-moderate	STSS	
Mahmoudian et al. (2013)
( n= 28)	NR	NR	Mostly in the head	No	3 and above	Chronic	THI and TQ	
Mannarelli et al. (2017)
( n= 20)	NR	NR	B/L	NR	6 and above	Chronic	THI	
Mohebbi et al. (2019)
( n= 20)	6-9 kHz	VAS
8.2 (1.23)	B/L	Not in the past 3 months	6 and above	Decompensated tinnitus	THI and TQ	
Rossiter, Stevens & Walker (2006)
( n= 18)	NR	NR	B/L	NR	3 and above	Moderate
tinnitus	TRQ	
Shiraishi et al. (1991)
( n= 20)	NR	NR	NR	NR	NR	NR	NR	
Stevens et al. (2007)
( n= 11)	NR	NR	B/L	NR	24 and above	Severe	TQ	
Trevis, McLachlan & Wilson (2016)
( n= 26)	NR	VAS
41.92 (22.18)	B/L	NR	3 and above	Chronic	TCSHQ & THI	
Waechter & Brännström (2015)
( n= 20)	NR	NR	B/L	Yes (8)	6 and above	40.05 (moderate)	TQ (6 months post testing)	
Notes.

B/L Bilateral

U/L Unilateral

NR Not Reported

dBSL decibel sensation Level

VAS Visual Analogue Scale

TQ Tinnitus Questionnaire

TCSHQ Tinnitus Case Sample History Questionnaire

STSS Subjective Tinnitus Severity Rating

TRQ Tinnitus Reaction Questionnaire

QIPA Tinnitus Psychological Impact Questionnaire

TSCHQ Tinnitus Sample Case History Questionnaire

THI Tinnitus Handicap Inventory

S-TQ Short version of Tinnitus Questionnaire

Hearing acuity

Hearing thresholds between the control and tinnitus group were matched in eight of the studies (Araneda, Deggouj & Renier, 2015; Attias et al., 1996; Attias et al., 1993; Hong et al., 2016; Mahmoudian et al., 2013; Mohebbi et al., 2019; Trevis, McLachlan & Wilson, 2016; Waechter & Brännström, 2015). Three of the studies had not performed any audiological testing to screen the participants hearing (Heeren et al., 2014; Jackson, Coyne & Clough, 2014; Rossiter, Stevens & Walker, 2006). Five studies (Andersson et al., 2000; Cuny et al., 2004; Mannarelli et al., 2017; Shiraishi et al., 1991; Stevens et al., 2007) had not matched the hearing ability of the participants. Detailed descriptions of the hearing characteristics of the participants included in the study are shown in Table 2.

Table 2 Detailed description of the hearing characteristics of participants in the included studies.

Studies	Hearing tested	PTA Results	Matching	Additional comments	
Andersson et al. (2000)	Yes	17 dBHL(11SD) at better frequency to 31 dBHL (27SD) at worst frequency	No	20 of 23 in tinnitus group –HL, 4 amongst using HA	
Araneda, Deggouj & Renier (2015)	Yes	<35 dBHL	Yes	NIL	
Attias et al. (1993)	Yes	Sloping loss	Yes	NIL	
Attias et al. (1996)	Yes	Sloping loss	Yes	NIL	
Cuny et al. (2004)	Yes	<10 dBHL till 2 kHz and 30 dBHL in later frequencies	No	NIL	
Heeren et al. (2014)	No	-	NA	Medical check by Physician in hearing disorder and had sufficient ability to follow instructions	
Hong et al. (2016)	Yes	<25 dBHL	NR (appears matched)	NIL	
Houdayer et al. (2015)	Yes	<15 dBHL	NR	5 individuals had hyperacusis	
Jackson, Coyne & Clough (2014)	No	-	NA	Comfortable conversing in a quiet environment	
Mahmoudian et al. (2013)	Yes	<20 dB till 2kHz & 40 dB from 4 kHz to 8 kHz	Yes	NIL	
Mannarelli et al. (2017)	Yes	<20 dB till 2 kHz & 30 dBHL in later frequencies	No	8 individuals’ HFHL	
Mohebbi et al. (2019)	Yes	<20 dB till 2kHz & 40 dB till 8 kHz	Yes	NIL	
Rossiter, Stevens & Walker (2006)	No	-	NA	1 participant in tinnitus group wore HA, 14 others self-report of mild to moderate HL	
Shiraishi et al. (1991)	Yes	Minimum 5.5 dBHL (9.16SD) @1 kHz to maximum of 22.08 dBHL (21.36SD) @ 8 kHz	No	Control audiogram data not available –stated as normal	
Stevens et al. (2007)	Yes	HFAHL 37.24 dBHL	No	TG- 8 HFHL (6 -moderate & 2 severe)
CG- 6 HFHL (5 mild & 1 profound)	
Trevis, McLachlan & Wilson (2016)	Yes	<25 dBHL	Yes	3 in CT group had HL (1 slight & 2 moderate)
Removal made no change, hence retained	
Waechter & Brännström (2015)	Yes	<20 dBHL	Yes	NIL	
Notes.

PTA Results Pure Tone Audiometric test results

HFAHL high frequency average hearing level (500, 1,000, 2,000 and 4,000 Hz)

dBHL decibel Hearing Level

SD Standard Deviation

HL Hearing loss

HA Hearing aids

kHz kiloHertz

HFHL High Frequency Hearing Loss

HFAHL High Frequency Average Hearing Level

Psychological and psychiatric factors

Nine of the studies (Andersson et al., 2000; Araneda, Deggouj & Renier, 2015; Heeren et al., 2014; Jackson, Coyne & Clough, 2014; Mannarelli et al., 2017; Rossiter, Stevens & Walker, 2006; Stevens et al., 2007; Trevis, McLachlan & Wilson, 2016; Waechter & Brännström, 2015) had screened for psychological factors such as anxiety and depression, while five did not (Cuny et al., 2004; Hong et al., 2016; Houdayer et al., 2015; Mahmoudian et al., 2013; Mohebbi et al., 2019). Attias et al. (1993) and Attias et al. (1996) based on interviews excluded individuals with psychological complaints. Shiraishi et al. (1991) reported undertaking psychological tests but had not reported the findings. Various questionnaires including Hospital Anxiety and Depression Scale (Snaith, 2003), State-Trait Anxiety Inventory (Spielberger, 1983), Beck’s Depression Inventory (Beck et al., 1961), Cognitive Failures Questionnaire (Broadbent et al., 1982), Mini-Mental State Examination (Folstein, Folstein & McHugh, 1975), Cornell Medical index test (Brodman et al., 1951), Zung depression questionnaire (Zung, 1965), Becks Anxiety Inventory (Beck & Steer, 1988), and Subjective Depression Rating Scale (Zung, Richards & Short, 1965) had been used to screen participants psychological status.

Behavioural and electrophysiological tests used and outcomes

The various tests carried out to assess attention with their major outcome are shown in Table 3.

Table 3 Tests and Outcome of various studies in the review.

Studies	Paradigm	Test done	Stimulus	Outcome	Study results –TG	
Andersson et al. (2000)	B	Stroop task	V	TG longer RT in classical and tinnitus word Stroop task	Executive function affected	
Araneda, Deggouj & Renier (2015)	B	Go/no-go task	A+V	TG slower RT and more false alarms in auditory modality	Cognitive inhibitory control mechanism affected	
Attias et al. (1993)	E	Oddball and Variable P300	A	TG P300 amplitude reduced, no changes in latency	Altered cognitive processing	
Attias et al. (1996)	E	Oddball P300	A+V	A: TG P300 prolonged and reduced amplitude
V: prolonged P3 TG	Selective attention affected	
Cuny et al. (2004)	B	Categorization task	A	Severe tinnitus performed less efficient than mild and moderate tinnitus	Disturbance in the automatic attention process	
Heeren et al. (2014)	B	ANT	V	TG longer RT. Alerting and orienting attention preserved with deficit in executive attention	A specific deficit in Top-down control and attention	
Hong et al. (2016)	E	Oddball P300	A	TG lower P300 amplitude	Impaired top-down attentional process	
Houdayer et al. (2015)	E	Oddball P300	A	No latency or amplitude difference in P300	Voluntary attention not affected	
Jackson, Coyne & Clough (2014)	B	Stroop task & Vienna Determination Test	V	TG longer RT, error rate no difference	Cognitive efficiency was affected.	
Mahmoudian et al. (2013)	E	MMN	A	TG lower amplitude and AUC for frequency, duration and SG deviants.	Pre- attentive sensory memory impaired	
Mannarelli et al. (2017)	E	Novelty P300	A	TG lower P300a amplitude, P300b comparable	A general slowing in the attentional switch to a salient stimulus	
Mohebbi et al. (2019)	E	MMN	A	Lower amplitude and AUC for high frequency and SG deviants in decompensated tinnitus.	A deficit in the pre-attentive change detection process	
Rossiter, Stevens & Walker (2006)	B	Reading span test & divided attention	A+V	TG lower reading span and longer RT category naming task	Controlled conscious cognitive process disrupted	
Shiraishi et al. (1991)	E	P300 & Contingent Negative Variation (CNV)	A+V	No latency or amplitude difference in P300	Comparable	
Stevens et al. (2007)	B	Stroop test & Visual divided attention	V	TG longer RT in word reading and category naming task	General degenerative effect on selective and divided attention	
Trevis, McLachlan & Wilson (2016)	B	Cognitive Control, Inhibition & Working Memory test	V	TG had Slow RT for cognitive control and inhibitory task	Reduced control to switch attention	
Waechter & Brännström (2015)	B	Modified Stroop task	V	No difference in RT and Accuracy	Results comparable	
Notes.

A Auditory

V Visual

A+V Auditory and Visual stimulus

E Electrophysiological paradigm

B Behavioural paradigm

MMN Mismatch Negativity

TG Tinnitus Group

AUC Area Under the Curve

SG Silent Gap

RT Reaction Time

Out of the seventeen studies, fourteen reported one or other forms of attention being affected in individuals with tinnitus.

Instrumentation

A total of eight studies had used electrophysiological measures. Four used a multichannel system with 29 to 32 channels (Hong et al., 2016; Houdayer et al., 2015; Mahmoudian et al., 2013; Mohebbi et al., 2019) and the remaining, between three and five channels. Stimuli used to elicit ERPs included Pure Tones, Tone Burst, a light flash and novel sounds. The intensity of the stimulus delivered was 50 decibels (Sound Pressure Level, SPL/Hearing Level, HL or Sensation Level, SL, elaborated in Table 4) and above. Houdayer et al. (2015) and Attias et al. (1996) have not provided intensity information. Most studies had used a simple oddball ratio of 80:20 except Mahmoudian et al. (2013) and Mohebbi et al. (2019) who used a 50:50 ratio with multiple deviants. Shiraishi et al. (1991) used an S1–S2 paradigm with 50:50 ratio.

Table 4 Stimulus and recording characteristics of electrophysiological studies.

Studies	Instrument –Recording	No. electrodes used	Ratio	Stimulus info	
Attias et al. (1996)	ORGIL BPM 30 system	5 electrodes
(Fz, Cz, Pz, T3 & T4)	80:20	1 kHz & 2 kHz PT	
Attias et al. (1993)	ORGIL BPM 30 system	3 electrodes
(Fz, Cz & Pz)	80:20	1 kHz & 2kHz PT at 40 dBSL	
Hong et al. (2016)	BrainAmp DC amp	32 electrodes
(10–10 system)	80:20	0.5 kHz & TP/8 kHz Pure tone at 50 dBSPL	
Houdayer et al. (2015)	Brainvision analyse 2.0	29 electrodes
(10–20 system)	80:20	1 kHz & 2 kHz PT	
Mahmoudian et al. (2013)	BRAIN QUICK LTM	29 scalp electrodes
(10–10 system)	50:50 (10% each deviant)	0.5 kHz, 1 kHz & 1.5 kHz PT at 65 dBSPL	
Mannarelli et al. (2017)	Miar Sirius EEG-EP Multifunction system	Multi- channel
(Frontal, central & parietal sites)
10–20 system	80:10:10	0.5 kHz, 1 kHz PT and novel sound at 80 dBSPL	
Mohebbi et al. (2019)	BRAIN QUICK LTM	29 scalp electrodes
(10–10 system)	50:50 (12.5% each deviant)	7.5 kHz, 8 kHz & 8.5 kHz PT, 85 dBSPL	
Shiraishi et al. (1991)	NR	3 electrodes Frontal, central & parietal sites	S1–S2 task (50:50)	Tone burst at 1 kHz at 85 dBHL & light flash	
Notes.

Hz Hertz

kHz kiloHertz

dBSPL decibel Sound Pressure Level

TP Tinnitus Pitch

PT Pure Tones

Meta-analysis of behavioural tests

The nine behavioural tests included in the papers had tested various forms of attention (selective attention, executive attention, divided attention, alerting and orienting) either directly or indirectly. Eight of the nine studies were included in the meta-analysis based on data availability. The mean, standard deviation (SD), the total number of participants in each group, SMD with 95% Confidence Interval are depicted in Fig. 2. The results of the meta-analysis indicated that individuals with tinnitus have difficulty (p < 0.001) in attentional tasks (Fig. 2).

Results from behavioural studies had indicated that individuals with tinnitus had altered inhibitory control and experienced cross-modal interference (Araneda, Deggouj & Renier, 2015), a specific deficit in executive attention (Heeren et al., 2014), a general disturbance in the automatic attentional process that prevents the deviant detection system from working (Cuny et al., 2004), controlled processing task affected (Rossiter, Stevens & Walker, 2006), a deficit in selective & divided attention (Stevens et al., 2007), poor executive performance (Jackson, Coyne & Clough, 2014) and/or reduced control inability to switch attention (Trevis, McLachlan & Wilson, 2016). The study by Waechter & Brännström (2015) was the only one to find no difference in the cognitive interference (using a modified Stroop paradigm) between individuals with tinnitus and control participants.

Figure 2 Meta-analysis on behavioural test of attention.

The figure indicates all the behavioral measures of attention in individuals with and without tinnitus. A random effects meta-analysis was performed using the standardized mean difference (SMD) of the reaction time obtained from various studies.

Subgroup analysis

Since hearing is a strong confounder, a subgroup analysis was performed on the behavioural studies used for meta-analysis. Out of the eight, three (Araneda, Deggouj & Renier, 2015; Trevis, McLachlan & Wilson, 2016; Waechter & Brännström, 2015) have matched for hearing, while the rest did not. A meta-analysis performed on studies which did not match the hearing of the participants resulted in a significant pooled estimate (p < 0.0001). However, when a meta-analysis was solely performed on three studies that matched for the hearing, it did not result in a significant estimate (p = 0.10). The meta-analysis for the studies that have matched and not matched for hearing are shown in Figs. 3 and 4 respectively. The results of the subgroup analysis conclude that when hearing is matched between the groups, attention is not necessarily affected in individuals with tinnitus. Hence, matching the groups based on hearing is very essential.

Figure 3 Subgroup analysis of attention in tinnitus with hearing matched studies.

A random effects meta-analysis of three studies that have matched for hearing between the tinnitus and control group. Standardized Mean Difference (SMD) of both the groups were used to check for the overall effect size.

Meta-analysis of electrophysiological studies

Out of the eight ERP studies, six measured P300 and two, the MMN. Both the MMN studies (Mahmoudian et al., 2013; Mohebbi et al., 2019) had reported an impaired pre-attentive sensory memory or change detection process in individuals with tinnitus. Due to the limited number of studies, a meta-analysis was not performed on MMN in tinnitus.

Of the six P300, four reported reduced P300 amplitude in individuals with tinnitus (Attias et al., 1996; Attias et al., 1993; Hong et al., 2016; Mannarelli et al., 2017); no difference was found in two of the studies (Houdayer et al., 2015; Shiraishi et al., 1991). The mean and SD of P300 amplitude were unavailable from three studies for meta-analysis (Attias et al., 1993; Hong et al., 2016; Mannarelli et al., 2017). By corresponding with the respective authors, the missing data were obtained for one study and the remaining, missing data (standard deviation) was derived by using the F-values and mean. A random-effects meta-analysis was performed with the mean difference of P300 amplitude between the control and tinnitus group with six studies. The results showed that the P300 amplitude was significantly reduced in individuals with tinnitus (Fig. 5). P300 amplitude is sensitive to resource allocation (Polich, 2007) and task difficulty. Justification for a reduction in P300 amplitude in the review articles incuded, depleted cognitive resources to focus on a task, abnormal information processing, improper resource allocation and alteration in the central predictive coding.

Figure 4 Subgroup analysis of attention in tinnitus with hearing unmatched studies.

A random effects meta-analysis of five studies those have not matched for hearing between the tinnitus and control group. Standardized Mean Difference (SMD) of both the groups were used to check for the overall effect size.

Figure 5 Meta-analysis of P300 amplitude.

A random effects meta-analysis of P300 amplitude between the control and tinnitus group. The mean difference (MD) between the P300 amplitude was used to check for the overall effect size.

With respect to P300 latency, two studies that reported the latency (Houdayer et al., 2015; Shiraishi et al., 1991) had suggested no difference in P300 latency between the tinnitus and the control groups. Only one of six studies had reported prolonged latencies in individuals with tinnitus (Attias et al., 1996). Due to lack of data availability, a meta-analysis on P300 latency was not performed.

Discussion

Attention and habituation are two intertwined domains proposed to play important roles in the perception and maintenance of tinnitus. The prevailing notion is that increased attention towards tinnitus prevents individuals from habituating to it. This review aimed to find out whether the attention and habituation processes were affected in individuals with tinnitus. To reduce heterogeneity, only studies containing continuous and subjective tinnitus were included. Concerning habituation, none of the studies screened qualified for the narrative synthesis. The main findings from the reviews are discussed in the following sections.

Place of study

The majority of studies were from Europe; few were from Asia. Studies of groups from several large populations (e.g., China, India, North and South America and Africa) were either not found or meet with the inclusion criteria. Although outcomes of attention assessment are likely to be similar in persons with tinnitus from a different population, it can’t be stated with certainty that culture does not play a role. Hence, a globally valid method for the assessment of attention in tinnitus is deemed useful.

Team

A multidisciplinary team including professionals from audiology, psychology, psychiatry, ENT, neurology, and engineering have collaborated in the majority of the studies. In general, studies based on psychological experiments have considered covariate analysis while occasionally ignoring hearing acuity. Since hearing is a major confounder, there is a need for cognitive psychologists and audiologists to work in close collaboration to design experimental methods for various tinnitus population.

Tinnitus characteristics

From this review, it is evident that tinnitus of greater than a moderate degree is associated with some amount of attention deficit. However, more studies are warranted to assess whether this deficit is linearly related to tinnitus severity. Apart from this, it was also noted that two scales THI (n = 7) and TQ (n = 6) were used predominantly. However, studies varied in the use of questionnaires to denote tinnitus characteristics. In addition, tinnitus characteristics such as tinnitus pitch and loudness, have not been reported in most of the studies (n = 11). It is felt that standardization of assessment protocols and reporting of results could overcome these problems.

Hearing

Hearing loss commonly accompanies tinnitus. Eight studies either included individuals with hearing loss and/or did not test their participants’ hearing thresholds. Peripheral hearing can solely influence auditory selective attention, by increasing the time to form auditory objects or switch attention rapidly (Shinn-Cunningham & Best, 2008). Hence, hearing loss is a strong confounder and controlling hearing between the tinnitus and control groups is essential to comment on the influence of tinnitus on an individual’s attentional abilities.

Psychological factors

Psychological factors such as anxiety and depression are often associated with tinnitus. These factors can influence an individual’s attentional abilities. Most of the studies have screened and excluded individuals with anxiety and/or depression (Araneda, Deggouj & Renier, 2015; Attias et al., 1996; Attias et al., 1993; Hong et al., 2016; Houdayer et al., 2015; Mahmoudian et al., 2013; Mohebbi et al., 2019) as they were considered as major confounders. Few studies have measured and just reported psychological disturbance using anxiety and/or depression scales (Andersson et al., 2000; Mannarelli et al., 2017; Stevens et al., 2007; Trevis, McLachlan & Wilson, 2016), while others considered it as a covariate (Heeren et al., 2014; Rossiter, Stevens & Walker, 2006) or matched the psychological status in tinnitus and control group (Jackson, Coyne & Clough, 2014; Waechter & Brännström, 2015). In general, it was observed that studies that employed an electrophysiological methodology just screened the psychological variables (Attias et al., 1996; Attias et al., 1993; Hong et al., 2016; Houdayer et al., 2015; Mahmoudian et al., 2013; Mohebbi et al., 2019). A correlation of such variables with the ERP results would provide better insight into how anxiety and depression are in individuals with tinnitus.

Generalizability of results

The sample size of the individual studies included in the review ranged from 11 to 33 participants per group (mean = 20.52). Further, most of the studies did not perform a power analysis. With a low sample size, the generalizability of the individual study results to the population of tinnitus becomes debatable. The current review pooled information from 531 participants to perform a meta-analysis. Therefore, the results of this review could stand as preliminary evidence for an attentional deficit in individuals with tinnitus. However, when the hearing between the groups is matched, attention was not necessarily affected.

Attention in tinnitus

Attention is a multifaceted process that requires coordination from bottom-up and top-down processes. Salient features of the stimulus guide the bottom-up attentional system through the process of sensory analysis and classification. The internal guidance system formed using prior knowledge, planning and the task goal guides the top-down attention that helps to selectively attend to a stimulus and form appropriate decisions (Katsuki & Constantinidis, 2014). Any deficit in one or both processes can hamper an individual’s attentional ability. In the case of tinnitus, bottom-up and/or top-down processing is believed to be impaired (Asadpour et al., 2018; Dos Santos Filha & Matas, 2010; Gabr, Abd El-Hay & Badawy, 2011; Hong et al., 2016; Richardson, 2018; Vasudevan, Palaniswamy & Balakrishnan, 2019; Wang et al., 2018) suggesting a possible dysfunction in the attentional system (Araneda et al., 2018; Cuny et al., 2004; Dornhoffer et al., 2006; Heeren et al., 2014; Li et al., 2016; Mahmoudian et al., 2013; Mannarelli et al., 2017; Milner et al., 2020; Mohamad, Hoare & Hall, 2016; Tegg-Quinn et al., 2016; Trevis, McLachlan & Wilson, 2016).

In the present review, various behavioural tests including Stroop task, inhibitory test, attentional network task, vigilance test, reading span task, categorization test and divided attention test have been used to study attention. Most of these studies have reported that one or other domains of attention are affected in individuals with tinnitus. The meta-analysis performed in the present review using eight behavioural studies has also indicated that individuals with tinnitus performed poorly at tasks evaluating attention.

With respect to electrophysiological studies, two studies using MMN have reported that passive attention or the pre-attentive change detection process was impaired in individuals with tinnitus. A meta-analysis on P300 amplitude supported a definite alteration in P3 amplitude in individuals having tinnitus indicating an alteration in their selective attention abilities. Due to the non-availability of data, a similar analysis on the P300 latency was not carried out.

In the current review, more than 90 per cent of the behavioural studies have agreed upon an attentional deficit in individuals with tinnitus. However, only 60 per cent of the electrophysiological studies (using P300 and MMN) have agreed upon the same. This could be attributed to the methodological differences and/or the fact that these behavioural studies did not assess the physiological process associated with attention. Therefore, performing both behavioural and electrophysiological measures on the same individual can give an insight into both the perceptual and physiological attentional changes associated with tinnitus. In addition, studies that differentially assess bottom-up and top-down attention are mandated. As stated, attention is a broad construct & specific forms of attention need to be probed separately concerning tinnitus. In addition, there is a lack of consistency in reporting the results of these studies, especially those published before 2000. A standardized protocol with appropriate tests to avoid confounders and to report results is needed to integrate the findings from various research.

The review pooled information from seventeen studies. It provides evidence on some form of attentional deficit being present in individuals with tinnitus. Studying various types of attention in each participant in the group is warranted to get a better insight into its differential impact. In addition, it can be deduced that attentional abilities tested using experimental tasks in controlled environments are affected in individuals with tinnitus. Nevertheless, testing attention in real-life situations using everyday tasks would be more appropriate to comment on the attentional abilities in the tinnitus population.

Habituation

Habituation as a phenomenon has not been studied extensively in individuals with tinnitus. However, improper or lack of habituation to the phantom sound had been proposed to be a major reason for the persistence/maintenance of tinnitus (Hallam, Rachman & Hinchcliffe, 1984; Jastreboff, 1990). A literature search on habituation in tinnitus resulted in a few articles, with inconsistent results. Due to the high risk of bias, many of these studies did not qualify for the narrative synthesis. Most studies on tinnitus in the literature have commented that habituation was affected (Cuny et al., 2004; Heeren et al., 2014; Mohebbi et al., 2019; Rossiter, Stevens & Walker, 2006; Stevens et al., 2007; Trevis, McLachlan & Wilson, 2016). However, they have not specifically measured it using behavioural or electrophysiological tests. A possible reason for the absence of evaluation is that tinnitus habituation is hard to test and/or there is a lack of standardized tests for its study (Uus.2016). The creation of a new paradigm or modification of existing paradigms is required to measure habituation in individuals with tinnitus.

Limitations of review

The most common limitation seen across tinnitus studies is the heterogeneity in participants and methods. It is often difficult to homogenize the groups with respect to tinnitus causes, onset, hearing acuity, psychological factors, tinnitus type and severity. Further, the low sample size of these studies makes it difficult to generalize the results. The review did not include studies that tested Contingent Negative Variation, CNV (Hoke et al., 1998; Kropp et al., 2012; Proefrock & Hoke, 1995) on tinnitus population, which could have given additional information on habituation in tinnitus. However, the current review tried to integrate the findings of each study to give a better insight into attention and habituation in tinnitus.

Future directions

It is recommended that future research employs longer tasks that require concentration instead of short intensive cognitive tasks. Ecologically valid assessment of attention in simulated real-world settings or use of ecological momentary assessment (EMA) in the real world, should be added to methods employed. EMAs offer ecologically valid measurements at the expense of control over the environment. Interaction overtime between attention, habituation and different environments may be a useful avenue for research (Deutsch & Piccirillo, 2020; Searchfield, 2014). Functional brain imaging to establish a link between inhibitory control and prefrontal cortical areas, exploring the interactions between top-down and bottom-up neurodynamic processing would all be useful additions to the field (Araneda, Deggouj & Renier, 2015; Hong et al., 2016).

The neural underpinnings of tinnitus are still debated. Until the neurophysiology of tinnitus and its physiological effects are understood, treatment should only address known contributors to tinnitus such as emotion and poor coping skills (Jackson, Coyne & Clough, 2014). This review suggests that attention is another contributor to tinnitus that warrants clinical research. Cognitive rehabilitation programs to help shift attention to a salient stimulus, a focus on executive control of attention and auditory training therapies may be effective in this regard (Mannarelli et al., 2017; Trevis, McLachlan & Wilson, 2016).

Conclusion

Attention is affected in individuals with tinnitus but the nature of any deficits and interaction are difficult to interpret due to the heterogeneity in methods and populations tested. With respect to habituation, there are very few studies to draw any conclusions. There is a need to carry out studies that assess more than a single type of attention and habituation in the same participant so that the actual relationship between the two domains could be studied.

Supplemental Information

Supplemental Information 1 PRISMA checklist

Click here for additional data file.

Supplemental Information 2 Raw data extraction data sheet

Click here for additional data file.

Supplemental Information 3 Risk of Bias Analysis results

Click here for additional data file.

Supplemental Information 4 Risk of Bias analysis ratings by two reviewers

Click here for additional data file.

Supplemental Information 5 Rationale and contribution of systematic review

Click here for additional data file.

Additional Information and Declarations

Competing Interests

Author Contributions

Data Availability

The authors declare there are no competing interests.

Harini Vasudevan conceived and designed the experiments, performed the experiments, analyzed the data, prepared figures and/or tables, authored or reviewed drafts of the paper, and approved the final draft.

Kanaka Ganapathy analyzed the data, prepared figures and/or tables, authored or reviewed drafts of the paper, and approved the final draft.

Hari Prakash Palaniswamy conceived and designed the experiments, analyzed the data, authored or reviewed drafts of the paper, and approved the final draft.

Grant Searchfield and Bellur Rajashekhar analyzed the data, authored or reviewed drafts of the paper, and approved the final draft.

The following information was supplied regarding data availability:

The raw data are available in the Supplementary File.

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
