# Peer review of "Systematic review and meta-analysis on the effect of continuous subjective tinnitus on attention and habituation"

_PeerJ, doi:10.7717/peerj.12340_

## Round 0.1 · original submission · Major Revisions

Dear author,

Please find enclosed the reviewers' comments on your paper.

·

Basic reporting

- Overall, the manuscript is well-structured and reads fluently, although some minor grammatical errors and inconsistencies (especially noticeable to me in the discussion section) might be resolved by an additional read-through.
- The introduction provides a very nice overview of the context in which the results of this systematic review should be interpreted. However, I feel that the paragraph on attention (starting at line 60) might benefit from a bit more detail, and some of the information regarding attention provided in the Discussion section (starting at line 361) might be better moved up to the Introduction. Additionally, I would like you to go slightly more in depth about why the MMN and P300 can be used as electrophysiological measures of attention, and what exactly the difference between both measures is. There should at least be an explanation of why a smaller P300 amplitude might be viewed as a deficit in attention to facilitate interpretation of the results.
- I agree with the comments of the editor that the exact contribution of this systematic review, in light of previously published reports, should be stated. Specifically, systematic reviews regarding the impact of tinnitus on cognitive performance (Clarke et al., Trends in Hearing, 2020) and late auditory evoked potentials (Cardon et al., Plos One, 2020) have been published recently. Both of these reviews reported results that are very comparable to the results of the current manuscript. This does not necessarily make the results of this review any less relevant, as the combination of behavioral and electrophysiological indices is an aspect that was not addressed by these earlier publications, but those existing reviews should at least be addressed and referred to in this manuscript.
- I did not find any figure legends (neither in the manuscript itself nor in the additional figure files). Is this correct? Additionally, please improve the resolution of Figure 1.
- No raw data are supplied. I agree with the authors that the original published articles function as the raw data. Nevertheless, it would be beneficial to provide access to the completed data extraction form, in which the authors have summarized all included papers, should anyone wish to check the validity of the meta-analysis results.
- A minor issue: I think the number of included records mentioned in the abstract should not include the duplicates, i.e. should be changed from 2271 to 1293.

Experimental design

- The research question is clear to me, although it could be stated more explicitly. In particular, it might be compelling to state how the present research benefits the existing literature, especially regarding two recently published systematic reviews in the field (see also my above comment in the Basic reporting section).
- My main concerns lie with the methodology of the meta-analyses. Almost no information is provided in the Methodology section. I understand from the Results section that you performed two random-effects meta-analyses; this should also be stated in the Methodology section (together with a rationale, cf. heterogeneity of the included papers). Did you conduct any post hoc analyses in order to explore possible outliers or influential studies? Were there any investigations into possible publication / reporting bias, explored via funnel plots and/or Egger’s regression tests? Ideally, a separate paragraph regarding data analysis addressing all of these questions would be added to the Methodology section.
- It seems like there is a very large number of papers rejected due to high risk of bias. Would it be possible to add more details about this rejection procedure to the manuscript? In particular, I would appreciate answers to the following questions: how many papers were rejected based on each of the stated categories of bias? How did you assess the risk of bias, e.g. were these decisions based on quantifiable criteria? If possible, I would suggest to add a table regarding risk of bias assessment to the manuscript.
- Differences between tinnitus groups and control groups, especially regarding age and hearing levels, could have a serious impact on the outcomes of the meta-analyses (both for behavioral and electrophysiological paradigms). Would it be possible to account for possible differences in the meta-analysis? Have you, for instance, considered conducting subgroup analyses to parse out possible effects of differences in age and/or hearing level on the overall results?

Validity of the findings

- As no raw data (in the form of the filled-out data extraction form) are supplied, it is not possible to check the validity of the meta-analysis results. From figure 3, it seems that the Hong 2016 study might be considered an outlier. Did you check for outliers or influential studies in your analyses? Could you check whether the overall effect would still be significant if this study is removed from the dataset?
- The discussion accurately states the relevant conclusions and provides a very good overview of any deficits in the current literature, such as the question whether there exists a (linear?) relationship between attentional deficits and tinnitus severity, and the lack of studies investigating behavioral and electrophysiological indices in the same sample.
- I do not quite understand the comments made in the paragraph ‘Generalizability of results’ (line 355-358). Are the systematic review and, especially, the meta-analyses presented in this paper not specifically performed to address this very issue? Can you add a comment on this question to this paragraph?

Additional comments

This review aimed to summarize all relevant literature on the roles of attention and habituation in tinnitus. Within the framework of a systematic review, the authors provide an overview of the existing literature on tinnitus and attention, and seek to confirm that attention is affected in individuals with tinnitus by performing two separate meta-analyses. Unfortunately, the literature on habituation and tinnitus proved too heterogeneous to arrive at any definitive conclusions.
The authors provide a satisfying overview of the current literature and accurately pinpoint the deficits in the field. To thoroughly underpin the conclusions provided in this paper, some important methodological concerns regarding the conduction of the meta-analyses and the assessment of risk of bias should be addressed.

Reviewer 2 ·

Basic reporting

See general comments

Experimental design

See general comments

Validity of the findings

See general comments

Additional comments

The current review presents a narrative as well as quantitative synthesis of studies focusing on the attention mechanism of tinnitus. This is much-needed work. However, there are some major limitations of the review in terms of how it is conducted as well as reported some of which are outlined below:

1) The method section does not provide all details including the method used for meta-analysis. This limits the reproducibility of this work. Authors should consider using a standardized reporting format.
2) There are many issues in the way inclusion/exclusion criteria have been applied in relation to registered protocol. For example, the preregistered protocol does not discuss excluding studies based on the risk of bias, although almost all studies on habituation have been excluded. Authors have used CASP rating although they had planned to use a different scale.
3) The results section is superficial with the limited synthesis of findings. The authors fail to consider study design when synthesizing the results and have treated all studies as same. Despite conducting the risk of bias using CASP, the details of ratings have not been provided.

In my opinion, the review is not ready for publication in the current form. I would have liked to have provided a more in-depth review if it was a matter of writing. However, it appears that much work is needed to redo some elements of the review in line with the pre-registered protocol. It is at least required to discuss some reasons for deviations from the protocol. Again, I do not believe that the authors could justify their choices for deviating from the protocol in all instances.

Reviewer 3 ·

Basic reporting

The manuscript uses mostly clear and unambiguous language. The title could be improved. Some terminology needs to be more clearly defined.

The introduction could be improved as outlined under General comments for the authors.

I think it would be easier for the reader if the manuscript used different levels of headings so that Method, Results and Discussion were more clearly defined. Some of the tables could be improved as outlined under General comments for the authors.

Tables 3 and 4. I am missing the actual results from these studies. It is not possible to follow how differences in outcome measures may have influenced the meta-analysis.

Experimental design

The manuscript lacks a clearly stated aim. I would have expected an aim that clearly indicates the effect studied an in what population. I would appreciate if the aim was stated according to PICO.

The rationale for the study could be improved as outlined under General comments for the authors.

Some additional information is needed in the method as outlined under General comments for the authors.

Validity of the findings

According to the authors this is the first systematic review and meta-analysis on the effect of tinnitus on attention and habituation. However, the rationale for the present study could be more clearly stated. A better argument is needed as to why this review is “essential”.

I am missing the actual results from these studies. It is not possible to follow how differences in outcome measures may have influenced the meta-analysis.

One of the conclusions (in abstract and L450-455) from this study is that single case studies are important to be able to study interaction between attention and habituation. It is unclear from the argument put forward how this will be possible, why this conclusion is drawn and based on which results reported in the present study. It would be easy to argue – based on the findings in the manuscript - that the effect of tinnitus on attention is more adequately studied using larger, closely matched cohorts.

Additional comments

This manuscript is reports on a systematic review and meta-analysis of studies that examine the effect of tinnitus on attention and habituation. Overall, the text is easy to follow but several clarifications are needed. Also, there are several problems with the current version. I refer to line (L) below.

Major issues:
The title is not very clear. I cannot deduce what effect is studied. This needs to be more clearly stated.

I think it would be easier for the reader if the manuscript used different levels of headings so that Method, Results and Discussion were more clearly defined.

The manuscript lacks a clearly stated aim. I would have expected an aim that clearly indicates the effect studied an in what population. I would appreciate if the aim was stated according to PICO.

One of the conclusions (in abstract and L450-455) from this study is that single case studies are important to be able to study interaction between attention and habituation. It is unclear from the argument put forward how this will be possible, why this conclusion is drawn and based on which results reported in the present study. It would be easy to argue – based on the findings in the manuscript - that the effect of tinnitus on attention is more adequately studied using larger, closely matched cohorts.

L58-59 and elsewhere. It is stated that “A contributing factor to tinnitus disorder may be the attention focused on tinnitus…”. I agree but this needs to be more clearly explained. I assume that the authors mean that a portion of available cognitive resources are required to suppress tinnitus leaving less resources to perform the task at hand. This needs more explanation since it is stated in the discussion (L299-300) that “The prevailing notion is that increased attention towards tinnitus prevents individuals from habituating to it.”. This is not clearly stated in the introduction.

L83-95. I believe that habituation needs fuller explanation. Habituation to external sound sources may be different from habituation to tinnitus. The latter may be influenced by many different factors. “sensory gating” needs to be defined. The link between habituation and electrophysiological measures needs to be more clearly explained and a rationale why the latter is a reasonable measure of habituation is needed.

L97-105. The rationale for the present study could be more clearly stated. A better argument is needed as to why this review is “essential”. The aim stated here is imprecise.

L153-160. It is unclear how the assessment of bias was converted in to different levels of risks. How was this rating made?

L163-176. More information on the meta-analysis made is needed. The information provided here and in the results is not sufficient to be able to replicate the study. A rationale is also needed for the parameters collected in the data extraction. It would be easier for the reader if the main outcome measures were defined here as well and not only in the results.

L199-239. Tinnitus characteristics, hearing acuity and psychological and psychiatric factors a mentioned here. However, the introduction provides little information or rationale why these factors are important. I think it would be valuable for the reader to have some information about these factors in the introduction since they are an important part of the discussion.

In the discussion, I am missing a discussion on the fact the level of task demands may influence the relation between tinnitus and attention (c.f., Lavie, 2005). Increasing task demands may reduce the influence tinnitus have on performance since more resources are required to be able to solve the task while decreased task demands may increase the influence of tinnitus on attention.

In the discussion, I am missing a brief discussion on the pros and cons with the own study design and how the study was conducted.

Minor issues:
L43. Check tempus.
L49-51. This is probably the most dominating cause but subjective tinnitus may have different causes.
L70 and elsewhere. I think it would be more appropriate to not abbreviate “don’t” but to write “do not”.
L84. “model” should probably read “models”.
L166. It is unclear what “residual inhibition information” means. Please clarify.
L318. The confounder hearing needs to be explained earlier in the manuscript. Why is it a major confounder?
L327-329. It is unclear why the assessment of tinnitus characteristics is important. Please clarify.
L333-335. I am missing reference to the studies by Frans Lin and colleagues.
Table 1. The order of the studies is unclear. Check that the studies are accurately referenced. E.g., Weachter et al. should probably read Waechter and Brännström.
Table 2. It is unclear how PTA and HFHL are defined.
Tables 3 and 4. I am missing the actual results from these studies. It is not possible to follow how differences in outcome measures may have influenced the meta-analysis.
Figure 1. It is unclear what “Abstracts identified from 0.5 databases” means. Please clarify.

---

## Round 0.2 · Minor Revisions

Thank you for the changes made. Your manuscript has been reviewed again and there are some additional changes needed.


Sincerely Gerhard Andersson

·

Basic reporting

I thank the authors for making these adjustments to the manuscript. For me, the introduction now contains the necessary information to interpret the results of the paper. I would still suggest to add figure legends, especially for the forest plots, e.g. to specify whether mean differences or standardized mean differences were used.

Experimental design

Many thanks for providing the data extraction form. However, after reading through this document carefully, it is still not possible for me to check the results of the meta-analysis, as it is not clear which data were extracted from the original papers. The data extraction form contains a lot of data that are not very relevant for the paper, but does not (unless I have missed it) contain a tab with the raw data that were used as input for the meta-analysis. I would urge the authors to provide these data and streamline the data extraction form. In its current form, I do not see how the provided data extraction form enables the reproducibility of the reported results.

The methodology concerning the conduction of the meta-analysis has been sufficiently expanded in the manuscript.

The authors have also provided the risk of bias analysis forms. However, only decisions from reviewer 1 and 2 are available. They state that twenty-six papers have been rejected due to high risk of bias. However, a remaining point of concern for me is that according to the ROB analysis table, a large number of these papers has been rejected NOT based on bias, but based on their relevance to the aim of the review or used methodology (reasons that are given in the column ‘Reason for rejection’). These are papers that, in fact, should have been rejected BEFORE risk of bias assessment (i.e. after full-text screening). I would like to ask the authors to alter their results section accordingly and update the PRISMA flowchart to reflect these alterations.

Validity of the findings

No further comments.

Reviewer 3 ·

Basic reporting

Please refer to “Additional comments” below.

Experimental design

Please refer to “Additional comments” below.

Validity of the findings

Please refer to “Additional comments” below.

Additional comments

This revised version of the manuscript is improved over the previous. Most of the previous concerns have been addressed adequately by the authors. I refer to line (L) below.

Two versions of the abstract are provided. One in the system and one in the actual text.

L75 and elsewhere. I still think it would be more appropriate to not abbreviate “don’t” and write “do not”.

L79-101. I still believe that habituation needs fuller explanation. Habituation to external sound sources may be different from habituation to tinnitus. The latter may be influenced by many different factors. “sensory gating” still needs to be defined. The link between habituation and electrophysiological measures needs to be more clearly explained and a rationale why the latter is a reasonable measure of habituation is needed.

L85. Consider removing “&” before “habituation”.

L103-110. I think that the argument that the review is essential is still not a strong argument since no actual new reasoning has been added.

L110-113. I still think that the aim could be more clearly stated. It is still unclear what the study population is. I assume adults.

L283. Can a paradigm where the deviant stimulus occurs equal amount of times as the standard be considered to be an oddball-paradigm? Could it be an explanation as to why the largest range is seen for that study (see Figure 5)?

L301. Add period at the end of the sentence.

L403-404. The last sentence seems to contradict what is stated on L375-380.

L424-426. Please clarify what “A meta-analysis performed” refers to. Present study or a previous?

L448-450. I am still not convinced that single case studies will provide much information.

L474. The abbreviation CNV is only mentioned in one of the tables prior to this occurrence. It would be easier for the reader to write out it here.

Tables 1-4. It is unclear what order the articles in the tables are presented. Please use alphabetical order or similar.

Table 1. B, U, L and R need explanation. From a perceptual perspective one decimal is sufficient for assessments of loudness.

Table 2. From a perceptual perspective one decimal is sufficient for PTA results. I assume that PTA here stands for pure tone audiogram and not pure tone average. HFAHL needs definition (calculated from what frequencies?). Should it be “slopping” hearing loss and not “sloping”?

---

## Round 0.3 · accepted · Accept

Thank you for the revision. I am now happy to forward your manuscript.

Sincerely Gerhard Andersson